# Rapid Support for Older Adults during the Initial Stages of the COVID-19 Pandemic: Results from a Geriatric Psychiatry Helpline

**DOI:** 10.3390/geriatrics6010030

**Published:** 2021-03-22

**Authors:** Anna-Sophia Wahl, Gloria Benson, Lucrezia Hausner, Sandra Schmitt, Annika Knoll, Adriana Ferretti-Bondy, Dimitri Hefter, Lutz Froelich

**Affiliations:** Department of Geriatric Psychiatry, Central Institute of Mental Health, Medical Faculty Mannheim, Heidelberg University, 68159 Mannheim, Germany; Gloria.Spielmann-Benson@zi-mannheim.de (G.B.); Lucrezia.Hausner@zi-mannheim.de (L.H.); Sandra.Schmitt@zi-mannheim.de (S.S.); Annika.Knoll@zi-mannheim.de (A.K.); Adriana.FerrettiBondy@zi-mannheim.de (A.F.-B.); Dimitri.Hefter@zi-mannheim.de (D.H.); Lutz.Froelich@zi-mannheim.de (L.F.)

**Keywords:** COVID-19, hotline, quarantine, psychological impact, elderly, anxiety, depression

## Abstract

Background. The COVID-19 pandemic and governmental lockdown measures disproportionally impact older adults. This study presents the results from a psychiatric helpline for older adults in Mannheim, Germany, during the lockdown, set up to provide information and psychosocial support. We aim to elucidate the needs of older adults, their reported changes, and the psychological impact during the initial stages of the health crisis. Methods: A total of 55 older adults called the psychiatric helpline between April and June 2020. Information on demographics, medical and psychiatric history. as well as changes in daily life due to the pandemic was collected anonymously. Mental health status was assessed using the 7-Item Hamilton Depression Rating Scale (HAMD-7) and the Hamilton Anxiety Rating Scale (HAM-A). Results: Most callers were women, older adults (*M* = 74.69 years), single, and retired. In total, 69% of callers reported new or an increase in psychiatric symptoms, with anxiety and depressive symptoms being the most common ones. Age was significantly negatively correlated to higher levels of anxiety and depression symptoms. Individuals with a previous diagnosis of a psychiatric disease reported significantly higher levels of depressive and anxiety symptoms than those without a diagnosis. Conclusion: In older adults, the perceived psychological impact of the COVID-19 crisis appears to ameliorate with age. Individuals with a history of psychiatric disease are most vulnerable to negative mental health outcomes. Rapid response in the form of a geriatric helpline is a useful initiative to support the psychosocial needs of older adults during a health crisis.

## 1. Introduction

The COVID-19 pandemic has caused an unpreceded disruption of daily life around the world. Older adults over the age of 65 are considered to be at higher risk of severe complications and mortality from COVID-19 and thus are disproportionally impacted [1,2,3,4,5].Worldwide governmental virus containment efforts include a variety of restriction measures from social distancing to stay-at-home orders. In Germany, the initial stage of the pandemic, or “first wave”, is considered to have lasted from March to June 2020 until the lifting of most restriction measures. The disruption of daily life through new imposed measures that increase isolation may have a stronger impact in older adults, which could in turn have deleterious effects on their mental health and wellbeing. Research focusing on the impact of the COVID-19 pandemic and efforts that address the needs of older adults remains very limited.

Social isolation in older adults has been associated with increased negative health outcomes such as depression and suicidality, as well as worse physical health [6,7,8,9]. Recent studies investigating the psychological impact of the COVID-19 restriction measures reveal the associations with poorer mental health outcomes [10,11,12] with a few focusing specifically on older adults [13,14,15]. Their findings suggest that older adults reported increased depression and anxiety, with older women experiencing more symptoms than older men. While a number of position papers have been published, there has been relatively little documentation of the impact of the pandemic in older adults and the support initiatives tailored to address the needs of the aging population.

Studies from previous epidemics and a few recent studies from the COVID-19 pandemic reveal that free anonymous helplines are a valuable way to offer timely access support services and information for help-seekers [16,17,18,19,20]. To this end, during the initial wave of the COVID-19 pandemic in Germany, we created a regional geriatric helpline designed to offer psychosocial support and assess the mental health impact of the health crisis among older adults. In this article, we have analyzed a cohort of callers from our geriatric helpline, to better characterize their needs and explore the effect of the pandemic on mental health in older adults. This paper aims to (1) provide the methods for implementing a rapid support helpline specialized for older adults, (2) characterize the callers as well as their needs and concerns, and (3) provide insight into the impact of the pandemic by assessing the associations between sociodemographic variables and their mental health status.

## 2. Materials and Methods

### 2.1. Establishment of a COVID-19 Helpline

The COVID-19 helpline for older adults (>65 years old) went live on 13 April 2020, 3 weeks after the first enforced Germany-wide lockdown (22 March 2020), at the Central Institute of Mental Health, Mannheim (CIMH), University of Heidelberg. The CIMH is the only psychiatric institution in the city of Mannheim and provides full sectorized care for older adults (20% of 300,000 inhabitants). The aim of the helpline was to provide (1) information about COVID-19 and protective measures, (2) psychiatric and psychological counseling, (3) acute psychotherapeutic interventions, and (4) information of local care providers for assistance with daily living activities. Before the establishment of the helpline, information on local care providers for older adults (e.g., support on groceries shopping and other errands) from different organizations (governmental and non-governmental) was collected. Three psychiatrists and three psychologists from the geriatric psychiatry department and memory clinic (with specialization in diagnosis and treatment of geriatric depression, anxiety, and dementia) hosted the helpline in two daily work shifts (8 a.m.–12 p.m. and 12–5 p.m.). All psychiatrists and psychologists answering the calls of the helpline where trained to ask and fill in a paper-based systematic questionnaire containing questions of demographic and social characteristics, medical history, as well as symptoms of mental health and the Hamilton Anxiety and Depression Scale. Cases were discussed at a weekly meeting where supervision was provided.

The helpline was announced in several local newspapers, radio stations, as well as advertised on the homepage of the city of Mannheim as “Corona-Crisis: Telephone helpline for seniors”. It was advertised that callers could anonymously share their needs and concerns and receive psychosocial support and referral to additional psychosocial or psychiatric services. Calls were not recorded; however, callers were initially told that the information provided would be documented anonymously and further analyzed in a scientific manner. Verbal consent was acquired.

A paper-based survey was developed to collect the following information: demographic, psychiatric history, current psychiatric treatment or psychotherapy experience, somatic diseases, number of medications taken, new or increase in psychiatric symptoms, number of daily social interactions before the pandemic and during the pandemic, living and financial situation, and requirements of daily support by others. The Charlson comorbidity index (CCI) was used to assess the age-comorbidity of somatic disease [21]. Mental health status was systematically assessed using the Hamilton depression and Hamilton anxiety scales as one-off assessment toward the end of the survey.

### 2.2. Mental Health Status

#### 2.2.1. Depression

The 7-Item Hamilton Depression Rating Scale (HAMD-7) [22] is commonly used in clinical and research settings to evaluate the presence and severity of depression symptoms. Seven items are assessed in a semistructured interview to assess depressed mood, guilt, anhedonia, psychic anxiety, loss of energy (fatigue), somatic anxiety, and suicidal ideation. This shortened version has been shown to be a valid and reliable instrument for screening depression [22]. The cutoffs for the different levels of depressive symptoms were set according to McIntyre [22]: Scoring below 4 indicated no depressive symptoms (level I). A score of 4–11 was considered mild (level II), a score of 12–20 moderate (level III), and a score >20 severe depression (level IV).

#### 2.2.2. Anxiety

The Hamilton Anxiety Rating Scale (HAM-A) 14-Item version was used to assess and quantify the presence and severity of anxiety symptoms [23]. The 14-Item version, a clinician-rated semistructured interview, shown to have a high reliability and concurrent validity, was used [24]. Higher scores indicated greater anxiety symptom severity [25]. According to Hamilton [23], a score of <17 indicates none to mild levels of anxiety (level I), a score of 18–24 moderate (level II), a score of 25–30 moderate to severe (level III), and a score >30 severe levels (level IV) of anxiety.

### 2.3. Data Analysis

Data were entered into electronic records for further analysis. Statistical analysis was performed with SPSS Version 24. The descriptive statistics of the sample were computed for the sociodemographics characteristics, consisting of frequencies and percentages for categorical values and mean and standard deviations (SD) for scale variables. For the mental health variables (anxiety and depression), skewness and kurtosis values were obtained, and the Shapiro–Wilk test was performed to assess normal distribution (unimodal, skewness < 1; Appendix A). Bivariate associations between mental health variables and age (continuous variable) were assessed via Pearson’s correlation coefficient *r.* Significant differences in mean level of mental health variables between categories of dichotomous sociodemographic variables were assessed via *t*-test. Ninety-five percent confidence intervals (95% CI) for Cohen’s d effect sizes were calculated. The associations between mental health variables and sociodemographic variables were assessed in the following groups: by living status (alone vs. not), previous psychiatric diagnosis (yes/no), current intake of psychopharmacological drugs, high risk for COVID-due to comorbidities, and high frequency of social contact (>6 social contacts per week vs. less than 6).

### 2.4. Ethics Approval and Consent to Participate

Informed consent was obtained from the subjects when calling our helpline to use data collected during the call for scientific purposes according to the statutes of the ethics commission of the Faculty of Behavioral and Cultural studies at the University of Heidelberg. Only when subjects had given verbal consent to participate did we collect their data and ask specific questions—also concerning their mental health status, including the Hamilton Depression and Anxiety Scale. Written consent could not be received as this was an anonymous helpline. We presented the study design and data acquisition to members of the ethics committee II at the medical faculty Mannheim, University of Heidelberg. Ethical approval was waived due to the anonymous design.

## 3. Results

A total of 55 older adults called the helpline for older adults during the initial stage of the pandemic (13 April–15 June). Of those, 53 of the callers completed the demographic survey, out of whom 44 completed the HAMD-7 and 35 the HAM-A Scales. We received a higher frequency of numbers of calls during the initial peak of the COVID-19 pandemic in Germany in April 2020 compared to end of May and June 2020 (Figure 1A). The reasons for calling the helpline varied, with most reported reasons being general information about coronavirus (60.7%), requesting health and psychological problems (32.1%), psychiatric consultation (26.8%), loneliness (21.4%), or requesting help with problems of daily function (7.1%) (Figure 1B). The average duration of each call was 39.56 min (SD = 22.93, with the shortest call lasting 9 min and the longest call requiring 120 min).

### 3.1. I Demographics and Medical History

In total, 53 callers completed the demographic survey. Sociodemographic characteristics are presented in Figure 1C. Most callers were women (85.5%), older adults (Age M = 74.69 SD = 8.32; range 59–98), single (50.0%), living in a city (76.9%), and retired (90.0%). Most of the calls, except for two, were regional (radius of 50–70 km around Mannheim). Almost half or the callers were living alone (49.0%), with the rest either living with their spouse or partner (40.8%) or with their children (12.2%), and one caller was living in a care home.

In total, 46 (88%) callers reported known co-morbidities, with cardiovascular (N = 24, 35%), chronic lung disease (N = 14, 20%), diabetes (N = 8, 12%), current or previous diagnosis of cancer (N = 12, 18%), and neurological diseases (N = 10, 15%) among the most common comorbidities (Table 1). The average Charlson Comorbidity Index (CCI) score was 4.02 (SD = 1.52; range 0–11). In total, 45% of respondents (N = 23) were considered at high risk for COVID-19 based on their comorbidities of either COPD, diabetes or cardiovascular disease. In terms of medication intake, 37% were prescribed with antihypertensive or heart drugs versus 12% of diabetes or COPD 20% medication. Most callers (72%) took more than one medication daily (Table 1).

In total, 38.5% of the callers reported a psychiatric history, out of which, depression (32%) and anxiety disorders (7.7%) were the most common diagnosis. Out of 52 respondents, 21% were currently seeing a psychiatric for treatment, and 31% of callers received psychotropic drugs (Table 1). The majority of callers had no previous psychotherapy experience (67.3%), and 11.5% of callers had had continuous psychotherapy for years (Table 1).

### 3.2. II Changes in Daily Life and Social Interaction

As expected, 52.8% of callers reported a decrease in their social contact since the COVID-19 crisis started, with 56.6% having fewer conversations with contacts and 79.2% reporting fewer visits from others, while only 5.7% of the callers reported an unclear support of daily living due to the COVID-19 crisis (Figure 2A).

When asked about their significant changes in daily life, 25% reported social isolation as a current problem in their daily life, while 9.6% reported problems accessing or shopping for groceries. Only one caller announced issues with personal hygiene, and two callers with mobility (Figure 2A). New financial difficulties due to the COVID-19 crisis were not an issue for most of the callers (89.8%, Figure 2A).

Although interactions with spouse and partners did not change because of social restrictions in the initial peak of the COVID-19 pandemic, callers reported having less direct, personal contact with all other relatives, such as children, grandchildren, parents or siblings (Figure 2B). Before the health crisis, 42% of callers had over 10 social interactions per week, which during the lockdown decreased to 1.8%. During the lockdown, the most common number of interactions was reduced to 2–5 (42.6%) and 0–1 (31%) per week (Figure 2C).

### 3.3. III Mental Health Status

Over 69% of callers reported new or an increase in psychiatric symptoms during this time (Figure 3A), with anxiety (47.2%), depressed mood (56.0%), sleep disturbances (30.2%), anhedonia (32.1%), restlessness/agitation (22.6%), changes in cognition (concentration difficulties) (15.4%), and suicidal thoughts (15.1%) among the most common reported symptoms.

Around 5% of callers reported loss of access to therapeutic resources (ambulatory psychotherapy, self-help group, and church meetings), while 13.2% of callers reported a loss of weekly activities through sports clubs or cultural associations (Figure 3A).

#### 3.3.1. Depression

A total of 44 callers completed the depression scale measured by the HAMD-7 and showed a mean score of 7.23 (SD = 5.77; range 0–20, Figure 3B, Appendix A). Regarding the subscales, the most reported symptoms were restlessness (M = 1.7), depressed mood (M = 1.45), and anhedonia or loss of interest (M = 1.14, Figure 3B). A total of 19 (42.2%) were considered to have mild levels of depressive symptoms (Score 4–11); 11 (24.4%) showed moderate levels (score 12–20), while we did not find participants with severe depressive symptoms (score > 20 according to (Brooks et al., 2020; di Santo et al., 2020; Soklaridis et al., 2020), Figure 3D). A third of participants did not report depressive symptoms (score < 4, Figure 3D).

#### 3.3.2. Anxiety

A total of 35 participants completed the anxiety scale measured by the HAM-A and showed a mean score of 14.03 (SD = 8.45; range 0–29, Figure 3C, Appendix A). The most reported symptoms were tension, depressed mood, anxious mood, fear, insomnia, and somatic muscular symptoms. Assessing the different anxiety levels as described (Brooks et al., 2020; di Santo et al., 2020; Soklaridis et al., 2020), 60.0% of callers were considered to have mild levels of anxiety (score < 17, Figure 3D), 28.6% (N = 10, score = 18–24) showed moderate, and 11.4% (N = 4, score = 25–30) moderate to severe levels of anxiety. No participants revealed severe anxiety symptoms (score > 30, Figure 3D).

### 3.4. IV Associations between Demographics Variables, Social Changes, and Mental Health Status during the Pandemic

Age was significantly negatively correlated to higher levels of anxiety (*r* = −0.340 *p* = 0.045) and depression symptoms (*r* = −0.293 *p* = 0.054, Table 2), showing that older individuals report fewer symptoms. Individuals prediagnosed with a psychiatric disease reported significantly higher levels of depressive (*t*(42) = 3.33, *p* = 0.001) and anxiety symptoms (*t*(33) = 4.00, *p* > 0.001) than those without a diagnosis. Likewise, those individuals who were currently taking psychiatric medication also reported significantly higher depressive (*t*(39) = 3.33, *p* = 0.002) and anxiety symptoms (*t*(31) = 2.97, *p* = 0.006). Individuals that lived alone reported significantly lower levels of symptoms of anxiety (*t*(33) = −2.57, *p* = 0.015), with no differences reported for depression symptoms (*p* = 0.017, Table 2). There were no significant differences in depression or anxiety symptoms for individuals who were either considered at high risk for COVID-19 due to high comorbidities or those engaging in frequent social contact.

### 3.5. V Help Provided

We provided information around COVID-19 and recommended measures to protect oneself to almost two thirds of the callers (58.8%) (Appendix A). In 7 cases (13.7%), we were also asked for our professional estimation of a somatic comorbidity and referred to a doctor with the suitable specialization. For 29.4% (N = 15), we recommended psychiatric counseling. Most callers (56.9%) required an open ear for the current situation and sorrows. In rare cases, we also provided practical help for daily living, social contacts, and contact to welfare (Appendix A).

In three quarters (74.5%) of all calls, no subsequent treatment was required. However, for three participants (5.9%), we provided further psychiatric counseling in the form of referral to psychiatric diagnostics and treatment. Seven callers (13.7%) received repetitive psychotherapeutic support with up to six subsequent telephone appointments. In one case (1.9%), our social worker was involved (Appendix A).

## 4. Discussion

As governments grapple with combatting the COVID-19 pandemic by implementing renewed restriction measures, our study elucidates the needs of older adults and the psychological impact during the initial stages of the health crisis. The long-lasting psychological consequences from this pandemic are largely unknown. However, our results from the geriatric helpline help to fill the gap by providing insight into the timely efforts which mental health institutions can implement to cater specifically to older adults.

Our results showed that most callers were in need of general information about COVID-19 measures; however, a significant portion required help with psychosocial problems such as loneliness, daily functioning, and new psychiatric symptoms, namely, anxiety and depressed mood. The length of the calls implies the resources and level of involvement needed per caller. The frequency of the calls at the beginning, and after changes in the restriction measures, highlights the unclarity of the restriction measures and the need for further clarification from the authorities.

Our helpline was able to rapidly respond to the specialized psychosocial needs during the crisis. Our specified support, in particular, psychiatric counseling and therapeutic talks, highlights the urgency of ameliorating the mental health impact of COVID-19 in real time among older adults. One third of callers were recommended psychiatric counseling, while 13% of callers were in need and given continuous psychotherapeutic support. Although we did not measure the effect of the helpline on mental health, we used the information gathered to explore the impact of the pandemic on these older adults. In a time where most mental health centers were closed and ambulatory therapy reduced its activity due to the lockdown measures, our results corroborated the need for the quick adaptation of mental health services and institutions. As shown in previous pandemics, ensuring the continuity of psychiatric support and providing easy access to geriatric mental health care in the absence of standard practices are essential responses to mitigating negative mental health outcomes [16,17,19].

Older age is considered an important risk factor for COVID-19, where older adults are disproportionately negatively impacted. However, our results reveal that the psychological impact appears to ameliorate with age, with older adults reporting fewer symptoms of anxiety and depression. This is consistent with recent results and previous literature which identifies older age as a protective factor in dealing with disasters or crisis [16,18]. In the context of COVID-19, older adults showed better emotional wellbeing and were less reactive to COVID-19 stressors than younger adults [16,18]. Similarly, an initial study in China revealed that young individuals were at higher risk of suffering from anxiety than older adults during the outbreak [16,18]. In line with disaster crisis literature, older victims previously showed lower anxiety, stress, and depression symptoms than younger individuals [16,18], with researchers attributing this to their greater life experience, crisis exposure or by having to face fewer responsibilities [26,27]. Further studies should longitudinally explore this protective effect of age and its consequences in the context of the COVID-19 pandemic.

The majority of our callers reported either new or an increase in psychiatric symptoms. This highlights the psychological impact of COVID-19 among older adult population and supports the need for targeted psychological intervention strategies for seniors. Our results show that those individuals prediagnosed with a psychiatric disorder and those taking psychotropic medication reported higher anxiety and depressive symptoms than those without. Our findings add to the growing body of emerging literature which indicates that individuals with current and/or past psychiatric disorders may be particularly vulnerable to the negative psychological sequalae of the pandemic [16,18,28,29]. In contrast, individuals that were considered at higher risk for COVID-19 due to their somatic comorbidities did not report significantly higher levels of anxiety or depressive symptoms. This finding suggests that unlike psychiatric comorbidities, somatic comorbidities may not play a significant role in mental health outcomes in older adults. However, this should be further studied in larger samples.

In accordance with other studies across Europe, most callers experienced changes in their daily life, with social isolation being considered a problem in a quarter of callers [26,27,30]. Interestingly, our results show that living alone proved to be a protective factor against anxiety symptoms, as individuals cohabitating with others reported higher levels of anxiety. This is in contrast with previous findings from general population surveys, which found cohabitation to be a protective factor against psychological suffering and negative mental health outcomes [16,18]. Contrary to expectations, our data revealed that older individuals living alone were less anxious and coped better. Older individuals living alone were perhaps better able to cope with the social distancing rules or “stay-at-home” orders since they were more used to living alone for longer. Given the scarce and mixed evidence to this date, this possible psychosocial resilience factor merits further attention, specifically among older adults.

The limitations of this study should be considered when interpreting the findings. First, these data were collected during the initial stages of the pandemic. Concerns, daily experiences, and mental health outcome may change and evolve as the outbreak evolves. Consequently, results should be interpreted with caution, and future works should explore these variables at later stages of the pandemic. Second, because the main objective of the helpline was the provision of anonymous psychosocial support with brief interventions, we have no feedback on caller satisfaction or the follow-up for the referrals. Further studies should investigate and systematically explore the efficacy and effect of teleservices intervention. Initial studies show promising results [26,27], with more intervention studies on their way [13,31,32,33]. Third, our sample was limited, as advertised, to cater to older adults over the age of 65; therefore, age-related associations and differences must be treated with caution. Finally, our sample may not generalize to other populations as it is entirely subject to each individual’s motivation to call the helpline.

## 5. Conclusions

The present study elucidated the specific needs of older individuals through the implementation of a geriatric helpline. The rapid adaptation of mental health institution resources through such initiatives are necessary to support psychosocial needs and psychological wellbeing of older adults during a health crisis. Considering the pandemic will most likely have a lasting effect, ongoing help initiatives and follow-up studies are warranted to evaluate and mitigate the negative health outcomes in the vulnerable aging population.

## Figures and Tables

**Figure 1 geriatrics-06-00030-f001:**
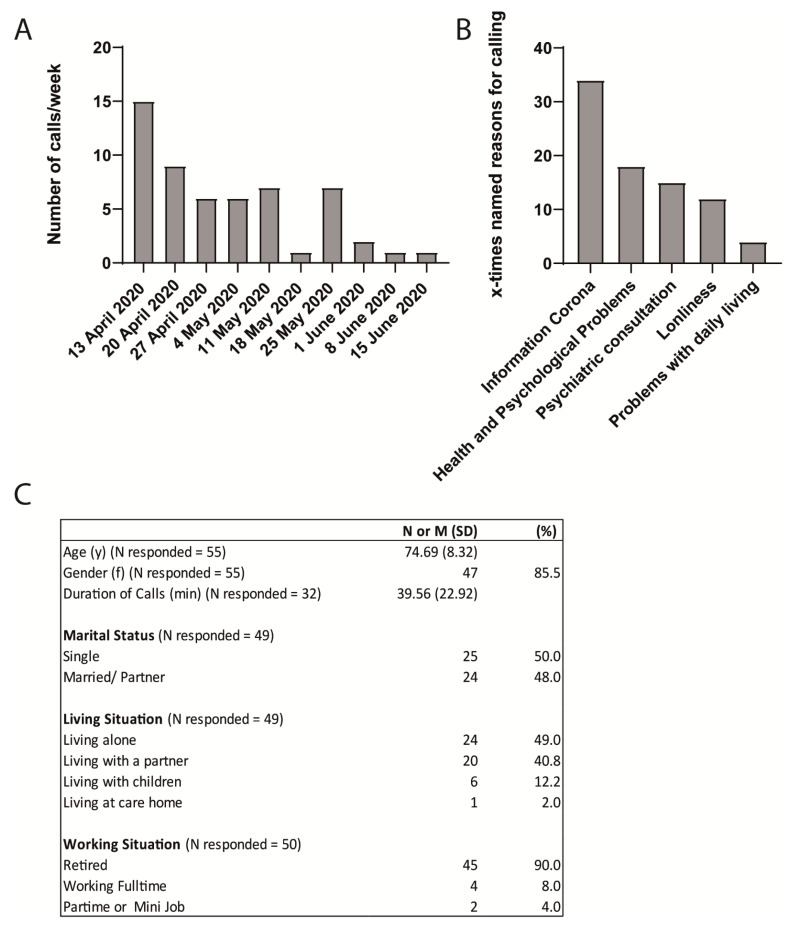
(**A**) Histogram showing the frequency of calls/week received at the geriatric helpline. (**B**) Graph depicting the reasons for calling the geriatric helpline. (**C**) Demographic characteristics of callers. N responded = the number of callers who were willing to answer the question. N represents the number of callers who affirmed the specific information and features asked. The percentage was calculated as (N/N responded) × 100.

**Figure 2 geriatrics-06-00030-f002:**
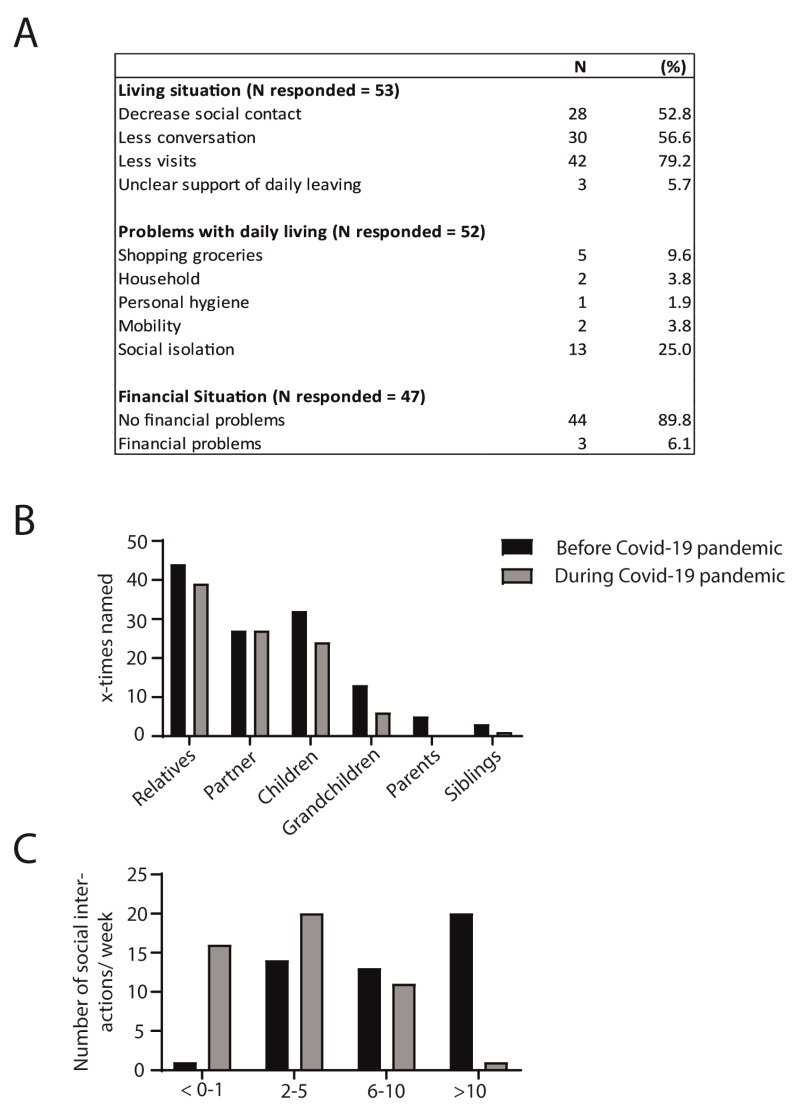
(**A**) Table depicting changes in daily living and in the financial situation due to the COVID-19 pandemic and lockdown measures. N responded = the number of callers who were willing to answer the question. N represents the number of callers who affirmed the specific information and features asked. (**B**) While the social interaction with one’s partner remained stable, callers reported fewer social interactions with all other relatives (including children, grandchildren, parents, and siblings) during the crisis. (**C**) The number of social interactions also decreased during the COVID-19 pandemic; while most participants had >10 social interactions in person per week before COVID-19, most callers reported a decrease in all personal social interactions to 2–5/week during the lockdown measures.

**Figure 3 geriatrics-06-00030-f003:**
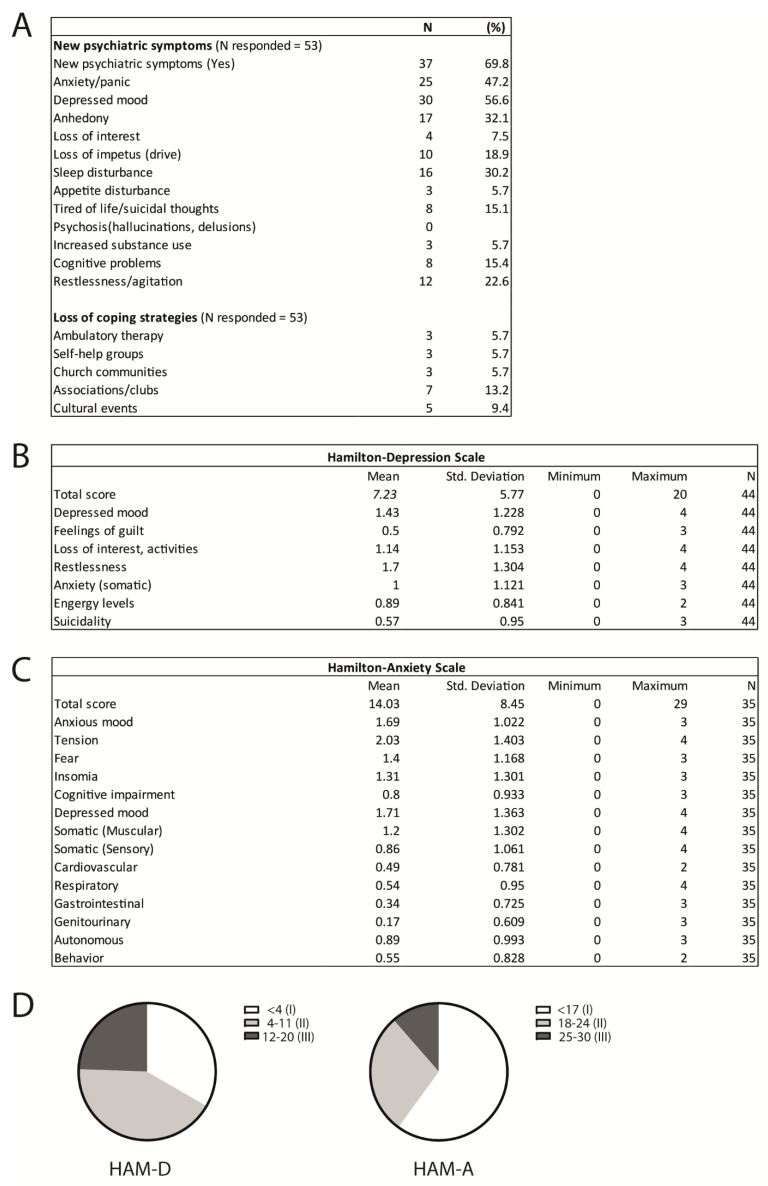
(**A**) Reported new psychiatric symptoms and loss of coping strategies during the COVID-19 pandemic. N responded = the number of callers who were willing to answer the question. N represents the number of callers who affirmed the specific information and features asked. The percentage was calculated as (N/N responded) * 100. (**B**) Mean scores and frequency of the Hamilton Depression Rating Scale (HAM-D) and (**C**) the Hamilton Anxiety Rating Scale (HAM-A) assessments. (**D**) Pie charts revealing the level of depressive or anxiety symptoms: A third (33.3%) showed no depressive symptoms (I) versus 42.2% with mild levels (II) and 24.4% with moderate levels (III). For the HAM-A, 60% of callers showed mild levels of anxiety (level I) versus 28.6% with moderate (II) and 11.4% with moderate to severe (III) levels of anxiety. We found neither severe (level IV) depressive nor anxious symptoms among the participants.

**Table 1 geriatrics-06-00030-t001:** Demographic and clinical characteristics assessing medical history and previous psychiatric and psychotherapy treatment and experience. N responded = the number of callers who were willing to answer the question. N represents the number of callers who affirmed the specific information and features asked. The percentage was calculated as (N/N responded) × 100.

	N or M (SD)	(%)
**Comorbidities (N responded = 52)**	46	88.5
Cardiovascular	24	46.2
Chronic lung disease	14	26.9
Diabetes	8	15.4
Current or previously diagnosed cancer	12	23.1
Neurological diseases	10	19.2
**Number of daily medications (N responded = 48)**		
Less than one	13	27.1
More than one	35	72.9
**Kind of daily medication (N responded = 48)**		
Psychiatric medication	15	31.3
Antihypertensive or cardiac medication	18	37.5
Antidiabetic medication	6	12.5
COPD or asthma medication	10	20.8
**Prediagnosed with psychiatric disease (N responded = 52)**	20	38.5
Anxiety	4	7.7
Depression	17	32.7
Psychosis	1	1.9
Addiction	4	7.7
**Psychiatric treatment (N responded = 52)**		
Currently seeing a psychiatrist	11	21.2
Previously seen a psychiatrist	5	9.6
Never in psychiatric treatment	33	67.3
**Previous Psychotherapy experience (N responded = 52)**		
No previous therapy	35	67.3
Consultation or less than 6 months	7	13.5
Psychotherapy more than 6 months	4	7.7
Continuous psychotherapy over years	6	11.5

**Table 2 geriatrics-06-00030-t002:** Associations between sociodemographic variables and mental health status. Differences in mean levels between categories were assessed via *t*-test. Individuals living alone reported higher anxiety symptoms than those living with others. Individuals with a prediagnosed psychiatric disorder reported more anxiety and depressive symptoms than callers without. Callers currently taking psychiatric medications reported significantly more depressive and anxiety symptoms. There were no differences among those at higher risk for COVID-19 or those engaging in frequent social interactions. Age was negatively correlated with higher levels of depressive and anxiety symptoms. Significant p values (*p* < 0.05) are shown in bold.

				*Depression*		
*Variables*	*N*	Mean	SD	*T*	Cohen’s d	*p*
**Living Alone**						
No	22	8.41	5.578	1.371	0.41	0.178
Yes	22	6.05	5.851			
**Previous diagnosis of psychiatric disease**						
No	26	5.00	4.964	−3.441	**1.05**	**0.001**
Yes	18	10.44	5.437			
**Psychiatric medication**						
No	26	5.58	4.933	−3.33	**1.08**	**0.002**
Yes	15	11.13	5.475			
**High frequency social contact**						
No	32	7.09	6.061	−0.584	0.20	0.563
Yes	11	8.27	4.798			
**COVID high-risk group**						
No	24	8.00	6.386	0.972	0.21	0.337
Yes	20	6.3	4.943			
			*r*			*p*
Age			−0.293			0.054
				***Anxiety***		
***Variables***	***N***	**Mean**	**SD**	***T***		***p***
**Living Alone**						
No	17	17.53	8.84	2.569	**0.87**	**0.015**
Yes	18	10.72	6.755			
**Previous diagnosis of psychiatric disease**						
No	20	9.9	7.174	−4.006	**1.37**	**>0.001**
Yes	15	19.53	6.854			
**Psychiatric medication**						
No	21	11.43	7.972	−2.97	**1.07**	**0.006**
Yes	12	19.67	7.075			
**High frequency social contact**						
No	27	13.07	8.185	−1.236	0.50	0.225
Yes	8	17.25	9.114			
**COVID high-risk group**						
No	19	13.63	9.552	−0.299	0.10	0.762
Yes	16	14.5	7.22			
			*r*			*p*
Age			−0.34			**0.045**

## Data Availability

All data are presented in the paper and the supplementary material. For details of the analysis with SPSS data can be made available on reasonable request.

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
