# Peer review of "Rapid Support for Older Adults during the Initial Stages of the COVID-19 Pandemic: Results from a Geriatric Psychiatry Helpline"

_geriatrics, 2021, doi:10.3390/geriatrics6010030_

Round 1
Reviewer 1 Report
Page 3, line 90 "toward" not towardss
Page 6, line 234 "referal" not "referall" (editors please check)
Table 2 when you have a mean of 5 or 8, please put 5.00 or 8.00
Tables with mean comparisons, please provide Cohen's d to show the effect size.
Author Response
We thank the editor and the reviewers for their helpful comments on the second version of our manuscript. In our revision, we have addressed the concerns of the reviewers and improved the manuscript accordingly. We hope that the final version will meet approval for publication. The new highlighted changes are in green.
Please find our responses below. For the convenience of the reviewers, we have highlighted in green all essential text modifications in the manuscript.
Reviewer #1
Comment #1 Page 3, line 90 "toward" not towards
Response: Adapted
Comment #2 Page 6, line 234 "referal" not "referall" (editors please check)
Response: Adapted to referral, editors mut check for preferred English.
Comment #3 Table 2 when you have a mean of 5 or 8, please put 5.00 or 8.00
Response: Adapted
Comment #4 Tables with mean comparisons, please provide Cohen's d to show the effect size.
Response: Adapted as requested, and added in the method section
“Ninety-five percent confidence intervals (95% CI) for Cohen's d effect sizes were calculated” see table 2 on Page 14.
Reviewer 2 Report
This study is a meaningful study for the mental health of the elderly in the COVID-19 pandemic. It is an honor to review such a meaningful paper.
The introduction suggests the need for COVID-19 and the mental health of the elderly, but why? There is no need for research on whether a helpline should be implemented and quickly supported.
The initial stages of the COVID-19 Pandemic defined in this study are unclear, and it is unclear whether this study applies only to the early stages of COVID-19.
This study was conducted over the phone, did you receive professional training? Please add information about it.
Please provide clear information by adding approval numbers for research ethics.
The reliability and validity of the research measurement tool are missing.
It is expected that the subjects of the study will have difficulty collecting data over the phone as an elderly person. What efforts have you made to solve this problem?
In addition, the minimum call for the survey is recorded as 9 minutes. It is difficult to clearly collect data for this study in 9 minutes. Therefore, it would be good to produce results again after cleaning.
As a result of the research, we presented too many paragraphs separately. It is deemed necessary to be reorganized so that readers can easily understand it by organizing it according to the research purpose.
Author Response
We thank the editor and the reviewers for their helpful comments on the second version of our manuscript. In our revision, we have addressed the concerns of the reviewers and improved the manuscript accordingly. We hope that the final version will meet approval for publication. The new highlighted changes are in green.
Please find our responses below. For the convenience of the reviewers, we have highlighted in green all essential text modifications in the manuscript.
Reviewer #2
Comment #0 This study is a meaningful study for the mental health of the elderly in the COVID-19 pandemic. It is an honor to review such a meaningful paper.
Response: We thank the reviewer for their careful consideration and meaningful suggestions.
Comment #1 The introduction suggests the need for COVID-19 and the mental health of the elderly, but why? There is no need for research on whether a helpline should be implemented and quickly supported.
Response: We thank this reviewer for the comment. As stated in the introduction Covid-19 is affecting daily life all around the world. As elderly (>65 years) are at higher risk for severe complications and mortality, they are disproportionally impacted. Governments around the world have in particular implemented lockdown measures and rules for social distancing to protect this group of the population. However, due to social distancing, isolation and less coping strategies with modern communication media and social media not only physical but also mental health of the elderly might be affected. We introduced a regional helpline to specifically serve the practical and psychological needs of the older population in a German city (Mannheim) during the first lockdown (March 22-June 2020) in Germany. As stated in the introduction the aim of the paper was not to study whether a helpline should be implemented and quickly supported. Instead, we aimed at analyzing a cohort of callers from our geriatric helpline, to better characterize their needs and explore the effect of the pandemic on mental health in older adults (see introduction page 2, line 57 ff). Furthermore, this paper aims to 1) provide the methods for implementing a rapid support helpline specialized for older adults, 2) characterize the callers as well as their needs and concerns 3) provide insight into the impact of the pandemic by assessing the associations between sociodemographic variables and their mental health status.
That easily accessible helplines are required to provide practical help, information and mental support- in particular for the older population- is undoubted. It was demanded early in popular and scientific literature (Jawaid, Science 2020; Meng et al., 2020; Office et al., 2020) and is reflected by the many helplines installed all over the world and the consumer demand (https://www.health.gov.au/contacts/older-persons-covid-19-support-line; https://www.thesilverline.org.uk/; https://www.nhs.uk/conditions/social-care-and-support-guide/help-from-social-services-and-charities/helplines-and-forums/; https://www.dublindiocese.ie/covid-19-support-line-for-older-people/; https://www.newstalk.com/news/covid-19-helpline-older-people-receives-1100-calls-one-day-991928; https://health.economictimes.indiatimes.com/news/policy/1077-will-be-delhis-covid-19-helpline-number-for-senior-citizens/75482614; https://www.aa.com.tr/en/latest-on-coronavirus-outbreak/turkey-s-covid-19-helpline-assisting-elderly/1785126).
Comment #2 The initial stages of the COVID-19 Pandemic defined in this study are unclear, and it is unclear whether this study applies only to the early stages of COVID-19.
Response: We thank the reviewer for the opportunity to clarify this point in the manuscript. The anonymous hotline was established during the first government lockdown measures of the pandemic, i.e. first wave of the outbreak. For clarity, we added the following in the additions in the introduction, methods and limitation sections.
“In Germany, the initial stage of the pandemic, or “first wave” is considered to have lasted from March to June 2020 until the lifting of most restriction measures” (Page 1)
“The COVID-19 helpline for older adults (>65 years old) went live on April 13th 2020, 3 weeks after the first enforced Germany-wide lock-down (March 22, 2020)” (Page 2)
“First, this data was collected during the initial stages of the pandemic. Concerns, daily experiences and mental health outcome may change and evolve as the outbreak evolves. Consequently, results should be taken with caution and future works should explore these variables at later stages of the pandemic.” (Page 8)
Comment #3 This study was conducted over the phone, did you receive professional training? Please add information about it.
Response: We thank the opportunity to clarify this point in the manuscript. All psychiatrist and psychologists which staffed the helpline are all specialized in treating geriatric patients and were properly trained before the establishment of the helpline. Cases were regularly discussed in a weekly meeting where the department head (LF) provided supervision for situations requiring a further psychiatric opinion.
We have included the following information as requested (page 2):
“All psychiatrists and psychologists answering the calls of the helpline where trained to ask and fill in a paper-based systematic questionnaire containing questions of demographic and social characteristics, medical history as well as symptoms of mental health and the Hamilton Anxiety and Depression Scale. Cases were discussed at a weekly meeting where supervision was provided.”
Comment #4 Please provide clear information by adding approval numbers for research ethics.
Response: We presented the study design to the members of the ethics committee II at the medical faculty Mannheim, University of Heidelberg. However, due to the anonymous nature of our hotline ethical approval was waived. When subjects called our helpline they were first informed that data might be collected and analyzed in an anonymous way for scientific purposes. When they gave verbal consent to participate, we collected their data and asked specific questions – also concerning their mental health status including the Hamilton Depression and Anxiety Scale.
To clarify we have added the following paragraph to the “Ethics approval and consent to participate” (page 4):
“Only when subjects gave verbal consent to participate we collected their data and asked specific questions – also concerning their mental health status including the Hamilton Depression and Anxiety Scale. Written consent could not be received as this was an anonymous helpline. We presented the study design and data acquisition to members of the ethics committee II at the medical faculty Mannheim, University of Heidelberg. Ethical approval was waived due to the anonymous design.”
Comment #5 The reliability and validity of the research measurement tool are missing.
Response: We have added the requested information in the method section (page 3):
HAM-D “ This shortened version been shown to be a valid and reliable instrument for screening depression (McIntyre et al., 2002).”
And for HAM-A “The 14-Items version, a clinician-rated semi-structured interview, shown to have a high reliability and concurrent validity, was used (Maier, 1988)”
Comment #6 It is expected that the subjects of the study will have difficulty collecting data over the phone as an elderly person. What efforts have you made to solve this problem?
Response: We thank for this hint. But to clarify this, callers were not asked to collect data over the phone. Instead, all psychiatrists and psychologists answering the calls of the helpline where trained to ask and fill in a paper-based systematic questionnaire containing questions of demographic and social characteristics, medical history as well as symptoms of mental health and the Hamilton Anxiety and Depression Scale.
On page 2 of the methods section “Establishment of a Covid-19 helpline” we have now stated:
“All psychiatrists and psychologists answering the calls of the helpline where trained to ask and fill in a paper-based systematic questionnaire containing questions of demographic and social characteristics, medical history as well as symptoms of mental health and the Hamilton Anxiety and Depression Scale.”
In addition, the minimum call for the survey is recorded as 9 minutes. It is difficult to clearly collect data for this study in 9 minutes. Therefore, it would be good to produce results again after cleaning.
Response: We thank for this comment, allow us to correct this misinterpretation: Calls took 39.56 mins on average (as stated on page 4, line 147) with the shortest call lasting 9 mins and the longest one 120 min (we have clarified this now in brackets).
“with the shortest call lasting 9 mins and the longest call requiring -120 mins)”
Additionally, some callers, as stated in the result section, only completed the demographic portion, for which 9 minutes would have been more than enough. The N, for each answered question is provided in the figures and legends.
As a result of the research, we presented too many paragraphs separately. It is deemed necessary to be reorganized so that readers can easily understand it by organizing it according to the research purpose.
Response: We thank the reviewer for this hint. We have restructured the results section and removed unnecessary headlines that might disrupt the train of reading. We have now concentrated on four subsections: I Demographics and medical history, II Changes in daily life and social interaction, III New Psychiatric Symptoms due to COVID-19 and IV Help provided.
This manuscript is a resubmission of an earlier submission. The following is a list of the peer review reports and author responses from that submission.
Round 1
Reviewer 1 Report
The research topic is excellent. It would be better if the authors revised the study design to measure the effect before and after the helpline consultation. The contents to be revised are as follows.
This study's strength is that a helpline for the elderly may be beneficial to the older adults' mental health. However, this study does not show the strength of the study.
Authors need to describe previous studies on helpline in the introduction session.
The aim described in line 65 seems the aims of the helpline, not the objectives of this study. What is the aim of this study? Did the authors investigate the helpline's effect on mental health? Did the authors measure anxiety and depression on a one-off basis using the helpline?
To demonstrate the study's strength, the authors should measure pre and post after the intervention (helplines) and describe the helpline's effectiveness as a result.
Who has received the call, and how did the authors survey the phone?
A total of 55 older adults called the COVID-19 Psychiatric hotline, but if the marriage status is 50 and 50 in C in figure 1, isn't it 100? The numbers do not match.
The author seems to have written the paper too quickly.
For the authors to emphasize the helpline in their conclusion, the authors also highlighted the helpline's importance based on previous studies in the introduction session.
Reviewer 2 Report
On the first page on the left for citation, it seems that only the first two authors were noted.
Keywords: I'd suggest adding COVID-19, elderly
Line 42 use "remains" as subject is singular
Line 49 use "COVID-19" rather than "COVID-10"
Lines 49,50 Some citations seem to be enclosed in more than one set of parentheses when only one is needed
Line 55 suggest "were least optimistic about the outcome"
Line 57 suggest "that older adults were not more willing to strictly adher to new rules"
Line 57 this sentence seems to be an implied double negative so perhaps it can be rewritten more clearly
Line 59 "implement" not implemented
Line 62 "restriction of visits to clinics and nursing homes"
Line 77 Generally, sentences should start with a word unless the number has more than two digits. In this case, I'd suggest to start with "Three".
Line 78 suggest "as we established"
Line 100 It is not totally clear if over the two months only 55 calls were received at all or if perhaps 100 were received but only 55 agreed to participate in the research. As it stands, it would appear that nearly all the callers agreed to participate in rather lengthy research.
Line 110 "assess" not asses (donkeys = asses)
Line 117 "assess" not asses
Line 254 "reported less symptoms"
Page 10 "Psychopharmaca treatment" rather than "Pychopharmaca treatment"
Line 272, just above it. r as - .340 (don't leave out the zero)
Line 278 "reported feeling significant relief because of"
Line 302 "impact of COVID-19 in real time"
Line 303 "were recommended"
Line 306 "corroborated the need"
Line 364 "lasting effects"
One limitation of the study is that it appears to have mostly involved urban residents. In this reviewer's case, we live in a rural area on a 20 acre farm. When the shutdown occurred in a city sixty miles away, one of our children decided that they wanted nothing to do with it and moved all eight children to our farm for nine months. I suspect that such movements went more from urban to rural sites than the opposite. That leaves open room for future research on the mental health of older adults in rural areas. I had forgotten how much harder it is to run a household of ten to 12 persons rather than just two persons. More food to buy, cook, clean up; more dishes to use, wash, put away; more laundry to collect, wash, dry, fold, put away; more cleaning in general, etc. It's also more expensive and hard on older folks with fixed incomes to start paying for household costs of 10-12 rather than just two. I wonder if the households with more than one person in this study might have involved additional grandchildren as well as a spouse. In our case, illness among two other grandchildren led my wife to move 500 miles away, leaving me to manage the grandchildren by myself for five of the nine months. I still had a regular job to do, though from home, which created conflicts between my work responsibilities and my grandparenting ones.
Older adults may have done better because they had less expected of them. If my wife and I had been 80 years old, perhaps the idea of moving the grandchildren to our place might not have been expected; but as we are younger, it seemed like an ok idea. So perhaps it's an issue of expectations placed on the younger older adults.